# PICKING DAISIES IN PRIVATE: FEDERATED LEARNING FROM SMALL DATASETS

## ABSTRACT

Federated learning allows multiple parties to collaboratively train a joint model without sharing local data. This enables applications of machine learning in settings of inherently distributed, undisclosable data such as in the medical domain. In practice, joint training is usually achieved by aggregating local models, for which local training objectives have to be in expectation similar to the joint (global) objective. Often, however, local datasets are so small that local objectives differ greatly from the global objective, resulting in federated learning to fail. We propose a novel approach that intertwines model *aggregations* with *permutations* of local models. The permutations expose each local model to a daisy chain of local datasets resulting in more efficient training in data-sparse domains. This enables training on extremely small local datasets, such as patient data across hospitals, while retaining the training efficiency and privacy benefits of federated learning.

## 1 INTRODUCTION

How can we learn *high quality* models when data is *inherently distributed* into *small parts* that cannot be shared or pooled, as we for example often encounter in the medical domain (Rieke et al., 2020)? Federated learning solves many but not all of these problems. While it can achieve good global models without disclosing any of the local data, it does require sufficient data to be available at each site in order for the locally trained models to achieve a minimum quality. In many relevant applications, this is not the case: in healthcare settings we often have as little as a few dozens of samples (Granlund et al., 2020; Su et al., 2021; Painter et al., 2020), but also domains where DL is generally regarded as highly successful, such as natural language processing and object detection often suffer from a lack of data (Liu et al., 2020; Kang et al., 2019).

In this paper, we present an elegant idea in which models are moved around iteratively and passed from client to client, thus forming a daisy-chain that the model traverses. This daisy-chaining allows us to learn from such small, distributed datasets simply by consecutively training the model with the data availalbe at each site. We should not do this naively, however, since it would not only lead to overfitting – a common problem in federated learning which can cause learning to diverge (Haddad-pour and Mahdavi, 2019) – but also violate privacy, since a client can infer from a model upon the data of the client it received it from (Shokri et al., 2017). To alleviate these issues, we propose an approach to combine daisy-chaining of local datasets with aggregation of models orchestrated by a coordinator, which we term federated daisy-chaining (FEDDC).

In a nutshell, in a daisy-chain round, local models are send to a coordinator and randomly redistributed to clients, without aggregation. Thereby, each individual model follows its own random daisy-chain of clients. In an aggregation round, models are aggregated and redistributed, as in standard federated learning. Our approach maintains privacy of local datasets, while it provably guarantees improvement of model quality of convex models with a suitable aggregation method which standard federated learning cannot. For non-convex models such as convolutional neural networks, it improves the performance upon the state-of-the-art on standard benchmark and medical datasets.

Formally, we show that FEDDC allows convergences on datasets so small that standard federated learning diverges by analyzing aggregation via the Radon point from a PAC-learning perspective. We substantiate this theoretical analysis by showing that FEDDC in practice matches the accuracy of a model trained on the full data of the SUSY binary classification dataset, beating standard federated

learning by a wide margin. In fact, FEDDC allows us to achieve optimal model quality with only 2 samples per client. In an extensive empirical evaluation, we then show that FEDDC outperforms vanilla federated learning (McMahan et al., 2017), naive daisy-chaining, and FedProx (Li et al., 2020a) on the benchmark dataset CIFAR10 (Krizhevsky, 2009), and more importantly on two real-world medical datasets.

In summary, our contributions are as follows.

- FEDDC, an elegant novel approach to federated learning from small datasets via a combination of daisy-chaining and aggregation,
- a theoretical guarantee that FEDDC improves models in terms of $\epsilon, \delta$-guarantees, which standard federated averaging can not,
- a thorough discussion of the privacy aspects and mitigations suitable for FEDDC, including an empirical evaluation of differentially private FEDDC, and
- an extensive set of experiments showing that FEDDC substantially improves model quality for small datasets, being able to train ResNet18 on a pneumonia dataset on as little as 8 samples per client.

## 2 RELATED WORK

Learning from small datasets is a well studied problem in machine learning. In the literature, we find among others general solutions, such as using simpler models, and transfer learning (Torrey and Shavlik, 2010), to more specialized ones, such as data augmentation (Ibrahim et al., 2021) and few-shot learning (Vinyals et al., 2016; Prabhu et al., 2019). In our scenario, however, data is abundant, but the problem is that the local datasets at each site are small and cannot be pooled.

Federated learning and its variants have been shown to learn from incomplete local data sources, e.g., non-iid label distributions (Li et al., 2020a; Wang et al., 2019) and differing feature distributions (Li et al., 2020b; Reisizadeh et al., 2020a), but were proven to fail in case of large gradient diversity (Haddadpour and Mahdavi, 2019) and too dissimilar label distribution (Marfoq et al., 2021). For very small datasets, local empirical distributions may vary greatly from the global data distribution—while the difference of empirical to true distribution decreases exponentially with the sample size (e.g., according to the Dvoretzky–Kiefer–Wolfowitz inequality), for small sample sizes the difference can be substantial, in particular if the data distribution differs from a Normal distribution (Kwak and Kim, 2017).

FedProx (Li et al., 2020a) is a variant of federated learning that is particularly suitable for tackling non-iid data distributions. It increases training stability by adding a momentum-like proximal term to the objective functions. This increase in stability, however, comes at the cost of not being privacy-preserving anymore (Rahman et al., 2021). We compare FEDDC to FedProx in Section 7.

We can reduce sample complexity by training networks only partially, e.g., by collaboratively training only a shared part of the model. This approach allows training client-specific models in the medical domain (Yang et al., 2021), but by design cannot train a global model. Kiss and Horvath (2021) propose a decentralized and communication-efficient variant of federated learning that migrates models over a decentralized network and stores incoming models locally at each client until sufficiently many models are collected on each client for an averaging step, similar to Gossip federated learing (Jelasity et al., 2005). The variant without averaging is similar to simple daisy-chaining which we compare to in Section 7. FEDDC is compatible with any aggregation operator, including the Radon point (Kamp et al., 2017) and the geometric median (Pillutla et al., 2019). It can also be straightforwardly combined with approaches to improve communication-efficiency, such as dynamic averaging (Kamp et al., 2018), and model quantization (Reisizadeh et al., 2020b).

## 3 PRELIMINARIES

We assume iterative learning algorithms (cf. Chp. 2.1.4 Kamp, 2019) $\mathcal{A} : \mathcal{X} \times \mathcal{Y} \times \mathcal{H} \to \mathcal{H}$ that update a model $h \in \mathcal{H}$ using a dataset $D \subset \mathcal{X} \times \mathcal{Y}$ from an input space $\mathcal{X}$ and output space $\mathcal{Y}$, i.e., $h_{t+1} = \mathcal{A}(D, h_t)$. Given a set of $m \in \mathbb{N}$ clients with local datasets $D^1, \ldots, D^m \subset \mathcal{X} \times \mathcal{Y}$ drawn

iid from a data distribution $\mathcal{D}$ and a loss function $\ell : \mathcal{Y} \times \mathcal{Y} \to \mathbb{R}$, the goal is to find a single model $h^* \in \mathcal{H}$ that minimizes the risk

$$\varepsilon(h) = \mathbb{E}_{(x,y) \sim \mathcal{D}} \left[ \ell(h(x), y) \right] . \tag{1}$$

In *centralized learning*, the datasets are pooled as $D = \bigcup_{i \in [m]} D^i$ and $\mathcal{A}$ is applied to $D$ until convergence. Note that applying $\mathcal{A}$ on $D$ can be the application to any random subset, e.g., as in mini-batch training, and convergence is measured in terms of low training loss, small gradient, or small deviation from previous iterate. In standard *federated learning* (McMahan et al., 2017), $\mathcal{A}$ is applied in parallel for $b \in \mathbb{N}$ rounds on each client locally to produce local models $h_1, \ldots, h_m$. These models are then centralized and aggregated using an aggregation operator $\mathrm{agg} : \mathcal{H}^m \to \mathcal{H}$, i.e., $\overline{h} = \mathrm{agg}(h_1, \ldots, h_m)$. The aggregated model $\overline{h}$ is then redistributed to local clients which perform another $b$ rounds of training using $\overline{h}$ as a starting point. This is iterated until convergence of $\overline{h}$. In the following section, we describe FEDDC.

## 4 METHOD

We propose federated daisy chaining as an extension to federated learning and hence assume a setup where we have $m$ clients and one designated coordinator node.[1] We provide pseudocode of our approach as Algorithm 1.

**The client**    Each client trains its local model in each round on local data (line 4), and sends its model to the coordinator every $b$ rounds for aggregation, where $b$ is the aggregation period, and every $d$ rounds for daisy chaining, where $d$ is the daisy-chaining period (line 6). This re-distribution of models results in each individual model following a daisy-chain of clients, training on each local dataset. Such a daisy-chain is interrupted by each aggregation round.

**The coordinator**    Upon receiving models (line 10), in a daisy-chaining round (line 11) the coordinator draws a random permutation $\pi$ of clients (line 12) and re-distributes the model of client $i$ to client $\pi(i)$ (line 13), while in an aggregation round (line 15), the coordinator instead aggregates all local models (line 16) and re-distributes the aggregate to all clients (line 17).

**Communication complexity**    Communication between clients and coordinator happens in $O\left(\frac{t_{max}}{d} + \frac{t_{max}}{b}\right)$ rounds, where $t_{max}$ is the overall number of rounds. Although inherently higher than in plain federated learning, the overall amount of communication in daisy chained federated learning is still low. In particular, in each communication round, each client sends and receives only a single model from the coordinator. The amount of communication per communication round is thus linear in the number of clients and model size, similar to federated averaging.

In the following section we show that the additional daisy-chaining rounds ensure convergence for small datasets in terms of PAC-like $\epsilon, \delta$-guarantees.

## 5 THEORY

Next, we theoretically analyze the key properties of FEDDC in terms of PAC-like $(\epsilon, \delta)$-guarantees. For that, we make the following assumption on the learning algorithm $\mathcal{A}$.

**Assumption 1** $((\epsilon, \delta)$-guarantees)**.** *The learning algorithm $\mathcal{A}$ applied on all datasets drawn iid from $\mathcal{D}$ of size $n \geq n_0 \in \mathbb{N}$ produces a model $h \in \mathcal{H}$ such that with probability $\delta \in (0, 1]$ it holds for $\epsilon > 0$ that*

$$\mathbb{P}\left(\varepsilon(h) > \epsilon\right) < \delta .$$

*The sample size $n_0$ is a monotone function in $\delta$ and $\epsilon$, i.e., for fixed $\epsilon$ $n_0$ is monotonically increasing with $\delta$ and for fixed $\delta$ it is monotonically decreasing with $\epsilon$ (note that typically $n_0$ is a polynomial in $\epsilon^{-1}$ and $\log(\delta^{-1})$).*

---

[1]This star-topology can be extended to hierarchical networks in a straight-forward manner. Federated learning can also be performed in a decentralized network via gossip algorithms (Jelasity et al., 2005)

---

**Algorithm 1** Federated Daisy-Chaining FEDDC

---

**Require:** daisy-chaining period $d$, aggregation period $b$, learning algorithm $\mathcal{A}$, aggregation operator agg, $m$ clients with local datasets $D^1, \ldots, D^m$

1: initialize local models $h_0^1, \ldots, h_0^m$
2: **at local client $i$ in round $t$**
3:     draw random set of samples $S$ from local dataset $D^i$
4:     $h_t^i \leftarrow \mathcal{A}(S, h_{t-1}^i)$
5:     **if** $t \% d = d - 1$ or $t \% b = b - 1$ **then**
6:         send $h_t^i$ to coordinator
7:     **end if**
8:
9: **at coordinator in round $t$**
10:     receive models $h_t^1, \ldots, h_t^m$
11:     **if** $t \% d = d - 1$ **then**
12:         draw permutation $\pi$ of [1,m] at random
13:         for all $i \in [m]$ send model $h_t^i$ to client $\pi(i)$
14:     **end if**
15:     **if** $t \% b = b - 1$ **then**
16:         $\overline{h}_t \leftarrow \text{agg}(h_t^1, \ldots, h_t^m)$
17:         send $\overline{h}_t$ to all clients
18:     **end if**
19:

---

Here $\varepsilon(h)$ is the risk defined in Equation 1. We will show that aggregation for small local datasets can diverge and that daisy-chaining can prevent this. For this, we analyze the development of $(\epsilon, \delta)$-guarantees on model quality when aggregating local models with and without daisy-chaining.

It is an open question how such an $(\epsilon, \delta)$-guarantee develops when averaging local models. Existing work analyzes convergence (Haddadpour and Mahdavi, 2019; Kamp et al., 2018) or regret (Kamp et al., 2014) and thus gives no generalization bound. Recent work on generalization bounds for federated averaging via the NTK-framework (Huang et al., 2021) is promising, but not directly compatible with daisy-chaining: the analysis of Huang et al. (2021) requires local datasets to be disjoint which would be violated by a daisy-chaining round. Using the Radon point (Radon, 1921) as aggregation operator, however, does permit analyzing the development of $(\epsilon, \delta)$-guarantees. In particular, it was shown that for fixed $\epsilon$ the probability of bad models is reduced doubly exponentially (Kamp et al., 2017) when we aggregate models using the (iterated) Radon point (Clarkson et al., 1996). Here, a Radon point of a set of points $S$ from a space $\mathcal{X}$ is—similar to the geometric median—a point in the convex hull of $S$ with a high centrality (more precisely, a Tukey depth (Tukey, 1975; Gilad-Bachrach et al., 2004) of at least 2). For a Radon point to exist, the size of $S$ has to be sufficiently large; the minimum size of $S \subset \mathcal{X}$ is denoted the Radon number of the space $\mathcal{X}$ and for $\mathcal{X} \subseteq \mathbb{R}^d$ the radon number is $d + 2$.

Let $r \in \mathbb{N}$ be the Radon number of $\mathcal{H}$, $\mathcal{A}$ be a learning algorithm as in assumption 1, and $\varepsilon$ be convex. Assume $m \geq r^h$ many clients with $h \in \mathbb{N}$. For $\epsilon > 0, \delta \in (0, 1]$ assume local datasets $D_1, \ldots, D_m$ of size larger than $n_0(\epsilon, \delta)$ drawn iid from $\mathcal{D}$, and $h_1, \ldots, h_m$ be local models trained on them using $\mathcal{A}$. Let $\mathfrak{r}_h$ be the iterated Radon point with $h$ iterations computed on the local models. Then it follows from Theorem 3 in Kamp et al. (2017) that for all $i \in [m]$ it holds that

$$\mathbb{P}\left(\varepsilon(\mathfrak{r}_h) > \epsilon\right) \leq \left(r\,\mathbb{P}\left(\varepsilon(h_i) > \epsilon\right)\right)^{2^h} \tag{2}$$

where the probability is over the random draws of local datasets. This implies that the iterated Radon point only improves over the local models if $\delta < r^{-1}$. Consequently, local models need to achieve a minimum quality for the federated learning system to converge.

**Corollary 2.** *Given a model space $\mathcal{H}$ with Radon number $r \in \mathbb{N}$, convex risk $\varepsilon$, and a learning algorithm $\mathcal{A}$ with sample size $n(\epsilon, \delta)$. Given $\epsilon > 0$ and any $h \in \mathbb{N}$, if local datasets $D_1, \ldots, D_m$ with $m \geq r^h$ are smaller than $n_0(\epsilon, r^{-1})$, then federated learning using the Radon point does not improve model quality in terms of $(\epsilon, \delta)$-guarantees.*

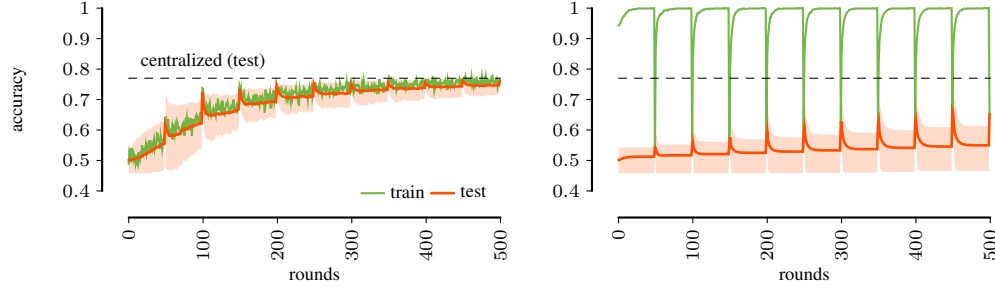

(a) FEDDC with $d = 1$, $b = 10$.          (b) Federated learning with Radon point with $b = 10$.

Figure 1: *Results on SUSY.* We visualize results in terms of train (green) and test error (orange) for FEDDC (a) and federated learning (b), both using Radon points for aggregation. The network has 441 clients with 2 data points per client. "Optimal performance, i.e., that of a central model trained on all data, is indicated by dashed line.

In other words, when using aggregation by Radon points alone, an improvement in terms of $(\epsilon, \delta)$-guarantees is strongly dependent on large enough local datasets. Furthermore, given $\delta > r^{-1}$, the guarantee can become arbitrarily bad by increasing the number of aggregation rounds.

Federated Daisy-Chaining as given in Algo. 1 permutes local models at random, which is in theory equivalent to permuting local datasets. This way, the amount of data visible to each model is increased. Since the permutation is drawn at random, the minimum amount of distinct local samples observed by each model can be given with high probability.

**Lemma 3.** *Given $\delta \in (0, 1]$, $m \in \mathbb{N}$ clients, and $k \in [m]$, if Algorithm 1 with daisy chaining period $d \in \mathbb{N}$ is run for $T \in \mathbb{N}$ rounds with*

$$T \geq d \frac{\ln \delta}{\ln \left( \frac{m-1}{m} \right) (m - k + 1)m}$$

*then each local model has seen at least $k$ distinct datasets with probability $1 - \delta$.*

*Proof.* For $m$ clients with $m$ local datasets, the chance of a client $i$ to not see dataset $j$ after $\tau$ many permutations is $\left( \frac{m-1}{m} \right)^{\tau}$. The probability that each of the $m$ clients is not seeing $m - k + 1$ other datasets is hence

$$\prod_{j=1}^{m-k+1} \left( \frac{m-1}{m} \right)^{\tau} = \left( \frac{m-1}{m} \right)^{\tau(m-k+1)} ,$$

and corresponds to the probability of each client seeing less than $k$ distinct other datasets. The probability of all clients seeing at least $k$ distinct datasets is hence at least

$$1 - \left( \frac{m-1}{m} \right)^{\tau(m-k+1)m} \overset{!}{\geq} 1 - \delta \Leftrightarrow \left( \frac{m-1}{m} \right)^{\tau(m-k+1)m} \overset{!}{\leq} \delta .$$

Taking the logarithm on both sides with base $(m - 1)/m < 1$ yields

$$\tau(m - k + 1)m \geq \frac{\ln \delta}{\ln \frac{m-1}{m}} .$$

Multiplying with $m - k + 1$ and observing that $\tau$ many daisy-chaining rounds with period $d$ require $T = \tau d$ total rounds yields the result. $\qquad\square$

From Lm. 3 it follows that when we perform daisy-chaining with $m$ clients, and local datasets of size $n$, for at least $d \ln \delta((\ln(m - 1) - ln(m))(m - k + 1)m)^{-1}$ rounds, each local model will with probability at least $1 - \delta$ be trained on at least $kn$ samples.

**Proposition 4.** *Given a model space $\mathcal{H}$ with Radon number $r \in \mathbb{N}$, convex risk $\varepsilon$, and a learning algorithm $\mathcal{A}$ with sample size $n(\epsilon, \delta)$. Given $\epsilon > 0$, $\delta \in (0, r^{-1})$ and any $h \in \mathbb{N}$, if local datasets $D_1, \ldots, D_m$ of size $n \in \mathbb{N}$ with $m \geq r^h$, then Alg. 1 using the Radon point with*

$$b \geq d \frac{\ln \delta}{\ln \left( \frac{m-1}{m} \right) \left( m - \frac{n_0(\epsilon, \delta)}{n} + 1 \right) m}$$

*improves model quality in terms of $(\epsilon, \delta)$-guarantees.*

*Proof.* The number of daisy-chaining rounds before computing a Radon point ensure that with probability $1 - \delta$ all local models are trained on at least $kn$ samples with $k = n_0(\epsilon, \delta)/n$, i.e., each model is trained on at least $n_0(\epsilon, \delta)$ samples and thus an $(\epsilon, \delta)$-guarantee holds for each model. Since $\delta < r^{-1}$, this guarantee is improved as detailed in Eq. (2). $\qquad\square$

To support this theoretical result, we compare FEDDC using the iterated Radon point with standard federated learning on the SUSY binary classification dataset (Baldi et al., 2014), training a linear model on $441$ clients with only $2$ samples per client. The results in Figure 1 show that after $500$ rounds FEDDC reached the test accuracy of a model that has been trained on the centralized dataset (ACC=0.77) beating federated learning by a large margin (ACC=0.65). Before further investigating FEDDC empirically in Section 7, we discuss the privacy-aspects of FEDDC in the following section.

## 6 PRIVACY

A major benefit of federated learning is that data remains undisclosed on the local clients and only model parameters are exchanged. It is, however, possible to infer upon local data given model parameters (Ma et al., 2020). In classical federated learning there are two types of attacks that would allow such inference: (i) an attacker intercepting the communication of a client with the coordinator obtaining model updates to infer upon the clients data, and (ii) a malicious coordinator obtaining models to infer upon the data of each client. A malicious client cannot learn about other clients data, since it only obtains the average of all local models. In federated daisy-chaining there is a third possible attack: (iii) a malicious client obtaining model updates from another client to infer upon its data.

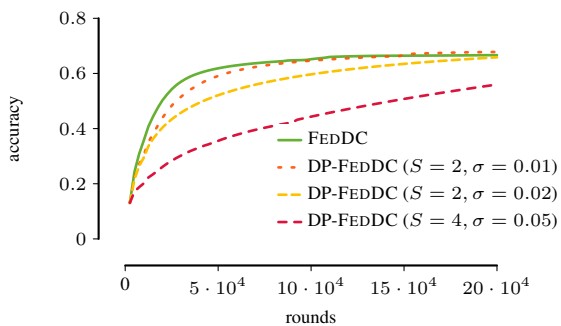

Figure 2: *Differential privacy results.* We show the performance of FEDDC (top solid line) compared to runs with clipped parameter updates and added Gaussian noise (dashed lines) on CIFAR10 with 250 clients.

In the following, we discuss potential defenses against these three types of attacks in more detail. Note that we limit the discussion on attacks that aim at inferring upon local data, thus breaching data privacy. For a discussion of attacks that aim to poison the learning process (Bhagoji et al., 2019) or create backdoors (Sun et al., 2019) for adversarial examples, we refer to Lyu et al. (2020).

A general and wide-spread approach to tackle all three possible attack types is to add noise to the model parameters before sending. Using appropriate clipping and noise, this guarantees $\epsilon, \delta$-differential privacy for local data (Wei et al., 2020) at the cost of a slight-to-moderate loss in model quality.

Another approach to tackle an attack on communication (i) is to use encrypted communication. One can also protect against a malicious coordinator (ii) by using homomorphic encryption that allows the coordinator to average models without decrypting them (Zhang et al., 2020). This, however, only works for particular aggregation operators and does not allow to perform daisy-chaining. Secure daisy-chaining in the presence of a malicious coordinator (ii) can, however, be performed using

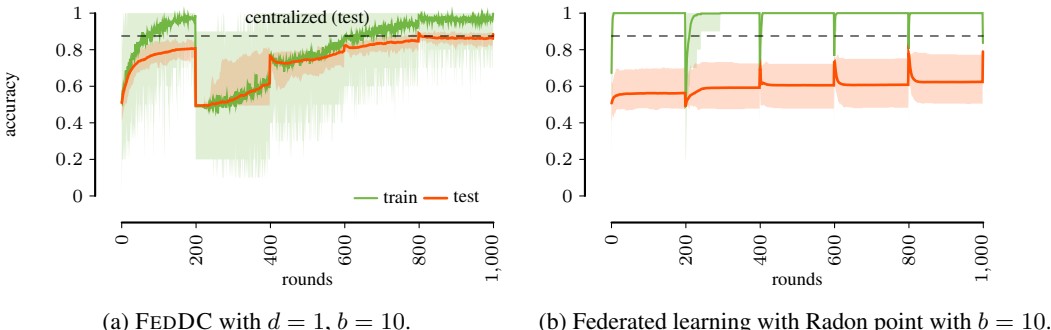

(a) FEDDC with $d = 1, b = 10$.                  (b) Federated learning with Radon point with $b = 10$.

Figure 3: *Synthetic data results.* Comparison of FEDDC (left) and FedAvg (right) for training MLPs on a synthetic dataset. Mean and confidence test accuracy of each client is reported in orange, where the optimal reachable accuracy, as given by centralized training, is indicated by the dashed black line.

asymmetric encryption. Assume each client creates a public-private key pair and shares the public key with the coordinator. To avoid the malicious coordinator to send clients its own public key and act as a man in the middle, public keys have to be announced (e.g., by broadcast). While this allows sending clients to identify the recipient of their model, no receiving client can identify the sender. Thus, inference on the origin of a model remains impossible. For a daisy-chaining round the coordinator sends the public key of the receiving client to the sending client, the sending client checks the validity of the key and sends an encrypted model to the coordinator which forwards it to the receiving client. Since only the receiving client can decrypt the model, the communication is secure.

In standard federated learning, a malicious client cannot infer upon the data of other clients from model updates, since it only receives the average model. In federated daisy-chaining, it receives the model from a random, unknown client in each daisy-chaining round. Now, the malicious client can infer upon the membership of a particular data point in the local dataset of the client the model originated from, i.e., a membership inference attack (Shokri et al., 2017). Similarly, the malicious client can infer upon the presence of data points with certain attributes in the dataset (Ateniese et al., 2015). The malicious client, however, does not know the client the model was trained on, i.e., it does not know the origin of the dataset. Using a random scheduling of daisy-chaining and averaging rounds at the coordinator, the malicious client cannot even distinguish between a model from another client or the average of all models. Nonetheless, daisy-chaining opens up new potential attack vectors (e.g., by clustering received models to potentially determine their origins). These potential attack vectors can be tackled by adding noise to model parameters as discussed above, since "[d]ifferentially private models are, by construction, secure against membership inference attacks" (Shokri et al., 2017). To investigate the impact of this privacy technique on FEDDC, we apply it in practice: We train a small ResNet on 250 clients using FEDDC with $d = 2$ and $b = 10$. Details on the experimental setup can be found in Supp. **??,??**. Differential privacy is achieved by clipping local model updates and adding Gaussian noise as proposed by Geyer et al. (2017). The results shown in Figure 2 indicate that the standard trade-off between model quality and privacy holds for FEDDC as well. Moreover, for mild privacy settings the model quality does not decrease. That is, FEDDC is able to robustly predict even under differential privacy.

## 7 EMPIRICAL EVALUATION

We evaluate FEDDC against the state-of-the-art in federated learning on synthetic and real world data. In particular, we compare to standard Federated averaging (FedAvg) (McMahan et al., 2017), FedAvg with equal communication as FEDDC, FedProx (Li et al., 2020a), and simple daisy-chaining without aggregation. As real world applications we consider the image classification problem CI-FAR10 (Krizhevsky, 2009), publicly available MRI scans for brain tumors[2], and chest X-rays for

---

[2]https://www.kaggle.com/navoneel/brain-mri-images-for-brain-tumor-detection

pneumonia (e.g., from COVID-19)[3]. For reproducibility, we provide details on architectures, and experimental setup in Supp. **??,??**. The implementation of the experiments is publicly available at `https://anonymous.4open.science/r/FedDC-1BC9`.

### 7.1 SYNTHETIC DATA

We first investigate the potential of FEDDC on a synthetic binary classification dataset generated by the sklearn (Pedregosa et al., 2011) `make_classification` function with 100 features. On this dataset, we train a simple MLP with 3 hidden layers on $m = 50$ clients with $n = 10$ samples per client. We compare FEDDC with $d = 1$ and $b = 200$ to FedAvg with $b = 200$. The results presented in Figure 3 show that FEDDC achieves an optimal test performance of $0.89$ (centralized training on all data achieves a test accuracy of $0.88$), substantially outperforming FedAvg. The results indicate that the main reason is overfitting of local clients, since for FedAvg train accuracy reaches $1.0$ quickly after each averaging step. In the following, we investigate how these promising results translate to real-world datasets.

### 7.2 CIFAR10

To compare FEDDC with the state of the art on real world data, we first consider the CIFAR10 image benchmark. To find a suitable aggregation period $b$ for FEDDC and FedAvg, we first run a search grid across periods for 250 clients with small versions of ResNet (details in Supp. **??**). We report the results in Figure 4 and set the period for FEDDC to $10$, and consider federated averaging with periods of both $1$ and $10$. For our next experiment, we equip 150 clients each with a ResNet18. To simulate our setting that each client has a small amount of samples, each one of them only receives 64 samples. Note that the combined amount of examples is only one fifth of the original training data, hence we cannot expect the typical performance on this dataset. As NNs are non-convex, Radon points are no longer suitable as aggre-

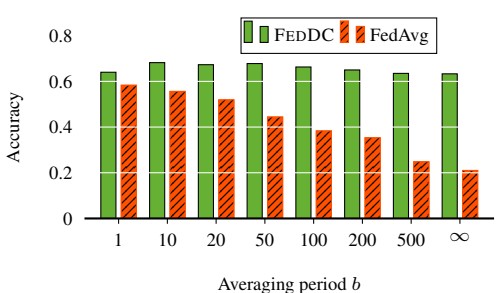

Figure 4: *Averaging periods on CIFAR10.* For 250 clients with small ResNets and 64 samples per client, we visualize the performance (higher is better) of FEDDC and FedAvg for different aggregation periods $b$.

gation method, we instead resort to averaging. Results are reported in Table 1. We observe that FEDDC achieves substantially higher accuracy of more than 6 percentage points over federated averaging with the same amount of communication. Looking closer, we see that FedAvg drastically overfits, achieving training accuracies of $0.97$, a similar trends as reported in Figure 3 for synthetic data. We further see that daisy-chaining alone, besides its privacy issues, performs worse than FEDDC. Similarly, FedProx run with $b = 10$ and $\mu = 0.1$ only achieves an accuracy of $0.545$.

| dataset | FedDC | Daisy-Chaining | FedAvg(b=10) | FedAvg(b=1) | FedProx |
|---|---|---|---|---|---|
| CIFAR10 | **62.8** | 59.2 | 51.0 | 56.3 | 54.5 |
| MRI | **78.4** | 57.7 | 75.6 | 74.1 | 76.5 |
| Pneumonia | **82.5** | 78.8 | 79.0 | 79.9 | 80.0 |

Table 1: Results on image data, reported are test accuracies of final model.

### 7.3 MEDICAL IMAGE DATA

We conduct experiments on real medical image data, which are naturally of small sample size and represent actual health related machine learning tasks. Here, we observe similar trends as for CIFAR10. For the brain MRI scans, we simulate 25 clients equipped with simple CNNs (see App. **??**) and 8 samples each. The results for brain tumor prediction based on these scans are reported in

---

[3]https://www.kaggle.com/praveengovi/coronahack-chest-xraydataset

Table 1. Again, FEDDC performs best, beating both FedAvg and FedProx on this challenging tasks. For pneumonia, we simulate 150 clients training ResNet18 (see again App. **??**) with 8 samples per client. The results in Table 1 not only show that FEDDC again outperforms all baselines, but also highlight that FEDDC enables us to train a ResNet18 to high accuracy with as little as 8 samples per client.

## 8 DISCUSSION

Empirical evaluation shows that FEDDC drastically improves upon state-of-the-art methods for federated learning for settings with only small amounts of available data. This confirms the theoretical potential, given by the $\epsilon, \delta$-guarantees, of improving model quality, which is unique among federated learning methods.

Using the iterated Radon point as aggregation method, and given as few as 2 samples per client, FEDDC matches the test accuracy of a model trained on the whole SUSY dataset, outperforming standard federated learning by over 12% points of accuracy. This result shows that unlike federated learning, FEDDC does not heavily overfit and is able to learn a generalized model, and is consistent with a synthetic prediction task using multi-layer perceptrons.

To study FEDDC in the context of real data, we consider both the standard image benchmark data CIFAR10, as well as two challenging image classification tasks from the health domain where only little data is available. On each of these tasks, FEDDC consistently outperforms state-of-the-art federate learning methods. Similar to before, we observe overfitting of standard federate learning methods. To rule out any effects due to increased communication, we also considered FedAvg with the same amount of communication as our method, however, FedAvg shows no improvement.

Through FEDDC, we present an effective solution to the problem of federated learning on small datasets. We further show that our method is able to robustly predict even under the effect of differential privacy, and suggest effective measures based on encryption as mitigations against attacks on communication or malicious coordinators.

## 9 CONCLUSION

We considered the problem of learning high quality models in settings where data is inherently distributed across sites, data cannot be shared between sites, and each site only has very little data available. We propose an elegant, surprisingly simple approach that effectively solves this problem, by combining the idea of model aggregation approaches from federated learning with the concept of passing individual models around while still maintaining privacy.

We showed that this approach theoretically improves models in terms of $\epsilon, \delta$-guarantees, which state-of-the-art federated averaging can not provide. In extensive empirical evaluations, including challenging image classification tasks from the health domain, we further show that for settings with limited data available per site, our method improves upon existing work by a wide margin. It thus paves the way for learning high quality models from small datasets.

Although the amount of communication is not a critical issue for the settings where we intend FEDDC to be used in, it does make for engaging future work to improve its communication efficiency and hence also enable it for settings with limited bandwidth, e.g., regarding model training on mobile devices. Both from a practical, as well as from a security and privacy perspective, it would also be interesting to study how to formulate FEDDC in a decentralized setting, when no coordinator is available.

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
