# OpenReview forum: "Picking Daisies in Private: Federated Learning from Small Datasets"
_ICLR.cc/2022/Conference — ICLR 2022 Submitted_

### Official Review · Reviewer_oYPW · 2021-10-30

**Correctness:** 3
**Technical Novelty And Significance:** 3
**Empirical Novelty And Significance:** 2
**Recommendation:** 5
**Confidence:** 4

**Main Review:**

Strengths:
- A novel extension to the FedAvg training procedure which includes model sharing between the clients and aggregation of models using iterated Radon point.
- Although the discussion on privacy concerns of the algorithm is general and may apply to other FL methods, I think that it is a nice addition to the paper.
- Most experimental details were given. Code was also provided (for the synthetic data experiment only though).

Weaknesses:
- The comparisons in the empirical evaluation are leaking. Most comparisons are to FedAvg which is rather old and generally isn't very strong. The authors keep stating that their method is state-of-the-art, yet it wasn't really compared against recent methods in order to consider it as such (for example, [1-4]).
- One of the main claims in favor of the proposed method is increased performance when local datasets are limited in size. Yet the paper misses important related work that addressed this issue as well (e.g., [1, 5]). I think that these studies and similar ones should be addressed in the revised version of the paper.
- Several alternatives were proposed for aggregating models instead of the standard model averaging (e.g., [6]). I believe the authors should address this line of research as well.
- The authors state that "The amount of communication per communication round is thus linear in the number of clients and model size, similar to federated averaging"; whilst the first part is true, I do not think that it is indeed similar to the communication cost of FedAvg which is sub-linear in the number of clients. At each communication round, FedDC communicates with all the clients while FedAvg (and similar methods) communicate with a small subset of them.
- In assumption 1, can you please clarify why for fixed $\epsilon$, $n_0$ is monotonically increasing with $\delta$?
- I think that background on the iterated Radon point method is missing. It cannot be expected from the average reader to be familiar with it.
- In section 7.2 it is stated the FedDC outperforms FedAvg even though they use the same amount of communication rounds. Can you please clarify how many communication rounds were used? I think that an analysis is missing here. Perhaps FedAvg was in overfit state which resulted in decreased generalization. I think that a better approach to compare between the models would be to declare a maximal number of communication rounds and use validation-based early stopping to select the best model.


[1] Achituve, I., Shamsian, A., Navon, A., Chechik, G., & Fetaya, E. (2021). Personalized Federated Learning with Gaussian Processes. arXiv preprint arXiv:2106.15482.
[2] Collins, L., Hassani, H., Mokhtari, A., & Shakkottai, S. (2021). Exploiting Shared Representations for Personalized Federated Learning. arXiv preprint arXiv:2102.07078.
[3] Shamsian, A., Navon, A., Fetaya, E., & Chechik, G. (2021). Personalized Federated Learning using Hypernetworks. arXiv preprint arXiv:2103.04628.
[4] Li, Q., He, B., & Song, D. (2021). Model-Contrastive Federated Learning. In Proceedings of the IEEE/CVF Conference on Computer Vision and Pattern Recognition (pp. 10713-10722).
[5] Hao, W., El-Khamy, M., Lee, J., Zhang, J., Liang, K. J., Chen, C., & Duke, L. C. (2021). Towards Fair Federated Learning with Zero-Shot Data Augmentation. In Proceedings of the IEEE/CVF Conference on Computer Vision and Pattern Recognition (pp. 3310-3319).
[6] Yurochkin, M., Agarwal, M., Ghosh, S., Greenewald, K., Hoang, N., & Khazaeni, Y. (2019, May). Bayesian nonparametric federated learning of neural networks. In International Conference on Machine Learning (pp. 7252-7261). PMLR.

**Summary Of The Paper:**

The paper presents a new training procedure for federated learning (FL) systems based on a daisy-chain network. Training the system has two phases, a daisy-chain phase in which models are transmitted from one client to another via a coordinator node, and the standard aggregation phase in which models are averaged according to FedAvg rule. The main motivation is reduced overfitting and improved generalization compared to the standard FedAvg training, especially with limited-sized local datasets. The paper presents PAC-like ($\epsilon, \delta$)-guarantees for their algorithm, and provides a discussion on privacy violation concerns of their algorithm. The method is demonstrated on several datasets and shows improved accuracy over the compared methods.

**Summary Of The Review:**

The paper presents a novel procedure for learning federated learning systems. I think that the paper misses important related works and that the empirical evaluation is not sufficient. Therefore, currently, I recommend rejecting the paper. I am willing to reevaluate the paper based on the author's response to my concerns.

---

### Official Review · Reviewer_hTem · 2021-11-03

**Correctness:** 3
**Technical Novelty And Significance:** 2
**Empirical Novelty And Significance:** 3
**Recommendation:** 5
**Confidence:** 4

**Main Review:**

Strengths:
+ To raise this real-world question is a shining point in this paper. To my best knowledge, there is limited work discussing this research question in this field;
+ They cover the key points in this paper including privacy discussion, communication cost discussion, and comparison with selected baselines.
+ They cite appropriate related works and the presentation of their work is easy to follow.

Weaknesses:
- As each local model will be re-distributed to other clients in this mechanism, though they discuss the privacy guarantee, it will raise security concerns of model poisoning and attack propagation in the network;
- According to the current random distribution mechanism, though authors find a substitution of sharing data by sharing models, the communication cost will increase a lot. The authors did propose the sparse matrix, but considering the whole iterative process in FL, the communication cost has not been discussed fully.
- The non-iid setting is another point that most research papers would discuss in their work. I am wondering how this approach would perform with this setting. Also, FedAvg and FedProx are both good baselines, but I am a little bit curious about the comparison with other state-of-the-art baselines.

**Summary Of The Paper:**

The authors propose a mechanism to solve the problem of small data in federated learning. Rather than directly uploading models to the server, there is an indicator to re-distribute each local model to other local models. They also discuss the conventional privacy guarantee with their approach. Finally, they provide experimental results on multiple datasets. The main contribution is to discuss how to solve the small data research question in FL. This question itself is practical and interesting. Also, I would say the analysis of the privacy guarantee is convincing.

**Summary Of The Review:**

The research question itself is an interesting topic to explore. The authors develop an algorithm to solve this question by exchanging model parameters across clients rather than exchanging data. This is good preliminary work in this direction. More work needs to be done in designing a novel coordinator, discussing the communication cost, and explorations on non-iid data settings.

---

### Official Review · Reviewer_pbKL · 2021-11-06

**Correctness:** 3
**Technical Novelty And Significance:** 2
**Empirical Novelty And Significance:** 2
**Recommendation:** 3
**Confidence:** 3

**Main Review:**

It is mentioned in the introduction that "While it can achieve good global models without disclosing any of the local data, it does require sufficient data to be available at each site in order for the locally trained models to achieve a minimum quality". I wonder if the authors can elaborate more on this by providing relevant references and a more detailed technical discussion.

A major concern in this work is the assumption of iid data. We know that in federated learning applications, the data is highly heterogenous. I wonder how well the proposed method would generalize to more realistic non-iid setting.

There is a subtle difference between the communication complexity of the two types of aggregation in the proposed method. When the coordinator aggregates the models and broadcasts the average (during aggregation period), there exist such broadcasting opportunity since all the clients receive the same model. However, during the daisy-chaining period, different models are pushed down to the clients, accounting for more communication complexity. Therefore, the characterization of communication complexity in Section 4 O(t_max/d + t_max/b) needs refinements.

It is not clear -in theory- how the performance of a federated learning algorithm is improved when using daisy-chaining technique. Proposition 4 is stated quite vaguely. Can the authors elaborate more on this?



**Summary Of The Paper:**

This paper considers a federated learning setting in which the sample size at client is so inadequet that the local objectives greatly differ from the global one. This paper proposes a novel approach that intertwines model aggregations with permutations of local models. By doing so, local models are exposed to several clients' data which ultimately improves the model accuracy.

**Summary Of The Review:**

In general, the technical and theoretical results of the paper are quite marginal. A clear discussion demonstrating the effect of daisy-chaining aggregate is missing. For instance, by using such method, how faster a federated learning algorithm e.g. FedAvg would converge.

Another concern is the assumption of iid data which makes the applicability of the proposed algorithm limited.

---

### Official Review · Reviewer_U9P6 · 2021-11-08

**Correctness:** 2
**Technical Novelty And Significance:** 2
**Empirical Novelty And Significance:** 2
**Recommendation:** 3
**Confidence:** 3

**Main Review:**

**Discalimer: I am not an expert on federated learning, hence I am not too knowledgeable about what has been proposed in the literature.**

The idea of increasing the sample diversity by swapping models is simple and natural. The paper deserves credit for proposing this (if indeed  this has not been considered before). On the other hand, the key downside to their approach is the privacy loss. This is what makes the federated model more challenging. In my view, this paper does not adequately analyze the privacy implications. A fair comparison to existing work in my view would look something like this:
1. Fix a privacy budget.
2. Prove a formal upper bound on the privacy loss incurred in the proposed model.
3. Repeat this analysis for state of the art federated learning, and compare the resulting accuracies.

In this work, step 2 is not addressed beyond experiments, and a quotation from Shokri etal, 2017.
In addition, there seem to be multiple issues with the way the paper is currently written, ranging from minor to fairly serious. In its current form, I cannot recommend the paper be published.

1. Assumption 1 feels a little ungainly, and circular. It feels like two things are being thrown together: one is a sample complexity bound for $H$, which is standard. The second is an assumption about the learning algorithm itself (which is what we would like to prove bounds about). Now some assumption on the algorithm is indeed called for, for instance if the aggregation function simply outputs the constant $0$ hypothesis, then the algorithm cannot work. But to sweep this under an assumption that the algorithm works properly given enough samples is somewhat tautological, it is definitely confusing. Perhaps the authors want to state an assumption on the aggregation function instead, which is indeed necessary and would not be circular.

2. Equation (2) quotes a result from the literature, given an upper bound on the error incurred by "radon point aggregation". It is conceivable that this can be improved, unless someone proves a lower bound  that  this is the best achievable rate. Perhaps the authors have such a bound in the supplementary material (which I am unable to see)? If not, Corollary 2 is not implied by Equation (2).   Even if such a lower bound is not known, it is fair to say that the this paper can lower the error on each individual model to be aggregated, so that Equation (2) is now more useful.

3. Lemma 3 is a simple corollary of the so-called coupon collector process.

4. For  fair comparison, the improved generalization needs to be balanced by a more formal analysis of the privacy loss. (Perhaps ideas from the shuffle model might be useful here. )






**Summary Of The Paper:**

This paper considers the setting of federated learning where each client holds a small subset $D_k$ of the entire dataset $D$. The goal is to learn a hypothesis with small (generalization) error. The challenge is that each dataset might be too small to guarantee convergence and/or generalization.

1. The  "obvious" solution to this is to have multiple clients share data. But this is undesirable because of privacy considerations.
2. Instead this paper proposes having clients swap "models" periodically. The paper argues that from the point of view of the model, it sees a greater diversity of client data, and thus ought to be able to train with better generalization. The paper reports that this improves over existing state of the art work in federated learning in terms of accuracy.
3. Of course this incurs increased privacy risk. The paper proposes to tackle this through differential privacy. The results reported here in this regard are experimental. The permutation is superficially similar to the recently proposed shuffle model, but no connection is drawn in the paper.



**Summary Of The Review:**


As it stands, the paper seems to benefit from the gains of data aggregation among random subsets of clients, without adequately grappling with the increased privacy risk. For this and the other reasons mentioned above, in my opinion this paper should not be accepted as it stands.

---

### Decision · Program_Chairs · 2022-01-20

**Decision:**

Reject

**Comment:**

The reviewers were not convinced by the authors' responses to their concerns, and this paper generated little followup discussion. Some primary concerns include the privacy analysis, limited technical contribution and scope (e.g., only being applicable to iid data), and lacking comparison to suggested baselines. The authors are suggested to take the reviewer comments into account for further investigation.